# Reduced replication but increased interferon resistance of SARS-CoV-2 Omicron BA.1

Rayhane Nchioua[1], Annika Schundner[2], Susanne Klute[1], Lennart Koepke[1], Maximilian Hirschenberger[1], Sabrina Noettger[1], Giorgio Fois[2], Fabian Zech[1], Alexander Graf[3], Stefan Krebs[3], Peter Braubach[5], Helmut Blum[3], Steffen Stenger[4], Dorota Kmiec[1], Manfred Frick[2], Frank Kirchhoff[1], Konstantin MJ Sparrer[1]

The IFN system constitutes a powerful antiviral defense machinery. Consequently, effective IFN responses protect against severe COVID-19 and exogenous IFNs inhibit SARS-CoV-2 in vitro. However, emerging SARS-CoV-2 variants of concern (VOCs) may have evolved reduced IFN sensitivity. Here, we determined differences in replication and IFN susceptibility of an early SARS-CoV-2 isolate (NL-02-2020) and the Alpha, Beta, Gamma, Delta, and Omicron VOCs in Calu-3 cells, iPSC-derived alveolar type-II cells (iAT2) and air–liquid interface (ALI) cultures of primary human airway epithelial cells. Our data show that Alpha, Beta, and Gamma replicated to similar levels as NL-02-2020. In comparison, Delta consistently yielded higher viral RNA levels, whereas Omicron was attenuated. All viruses were inhibited by type-I, -II, and -III IFNs, albeit to varying extend. Overall, Alpha was slightly less sensitive to IFNs than NL-02-2020, whereas Beta, Gamma, and Delta remained fully sensitive. Strikingly, Omicron BA.1 was least restricted by exogenous IFNs in all cell models. Our results suggest that enhanced innate immune evasion rather than higher replication capacity contributed to the effective spread of Omicron BA.1.

## Introduction

The IFN system is a powerful barrier against viral infections (Samuel, 2001; Stetson & Medzhitov, 2006; Sparrer & Gack, 2015). After recognition of viral pathogen-associated molecular patterns by germline-encoded pattern recognition receptors, signaling cascades are activated. This results in the induction and secretion of IFNs and other pro-inflammatory cytokines (Sparrer & Gack, 2015). IFNs are classified into three major types (I, II, and III) based on their receptor usage (Platanias, 2005). Type I IFNs include IFN-α and IFN-β. The sole type II IFN is IFN-γ and the four type III IFNs are IFN-λ1–4. Upon binding to their respective receptors, the expression of hundreds of so-called IFN-stimulated genes is induced. Among them are many antiviral effectors (Kluge et al, 2015; Schoggins, 2019), which set cells in an antiviral state and restrict the spread of pathogens, such as SARS-CoV-2 (Bastard et al, 2020; Zhang et al, 2020; Lopez et al, 2021; Sposito et al, 2021 Preprint; Zanoni, 2021). Although SARS-CoV-2 uses numerous immune evasion mechanisms (Sa Ribero et al, 2020; Thoms et al, 2020; Xia et al, 2020; Hayn et al, 2021; Lee et al, 2022), early and effective induction of IFNs in the lung prevents severe COVID-19 (Sposito et al, 2021 Preprint; Zanoni, 2021). Conversely, inborn defects in the IFN system or auto-antibodies against type I IFNs are frequently associated with severe COVID-19 (Bastard et al, 2020; Zhang et al, 2020).

The rapid worldwide spread of SARS-CoV-2 was associated with the emergence of variants that may show an increased ability to avoid immune control (Harvey et al, 2021; Planas et al, 2021; Thorne et al, 2022) and are called variants of concern (VOC). Currently, the five VOCs recognized by the World Health Organization in order of their appearance are: B.1.351, B.1.1.7, P.1, B.1.617.2, and B.1.1.529. For simplification, these are also referred to as Beta, Alpha, Gamma, Delta, and Omicron variants, respectively. While the population spread of the Beta and Gamma variants was limited to certain regions, the Alpha variant rapidly overtook other strains in most countries in early 2021 but was outcompeted by the Delta VOC in November 2021. By late 2021, the first subvariant of Omicron, BA.1, rapidly outcompeted Delta (Jung et al, 2022). Currently (March 2023), the Omicron variant still dominates the pandemic, with XBB.1.5 and BA.2.75 as the most prevalent subvariants. One hallmark of VOCs, especially the Omicron variant, is their enhanced escape from neutralizing antibodies (Cao et al, 2022; Pastorio et al, 2022; Planas et al, 2022). However, increased resistance towards innate immune defenses may also play a key role in its success and is currently debated (Bojkova et al, 2022b; Shalamova et al, 2022). To address this in physiologically relevant settings, we determined the IFN susceptibility of an early SARS-CoV-2 isolate from February 2020

[1]Institute of Molecular Virology, Ulm University Medical Center, Ulm, Germany   [2]Institute of General Physiology, Ulm University Medical Center, Ulm, Germany   [3]Laboratory for Functional Genome Analysis, Gene Center, LMU Munich, Munich, Germany   [4]Institute for Medical Microbiology and Hygiene, Ulm University Medical Center, Ulm, Germany   [5]Hannover Medical School, Institute for Pathology, Hannover, Germany

Correspondence: Konstantin.Sparrer@uni-ulm.de; Frank.Kirchhoff@uni-ulm.de

(NL-02-2020) and five VOC isolates, including Omicron BA.1, in three different human lung cell models.

## Results

To confirm the accuracy of the viruses used in this study, we previously performed next-generation sequencing (Nchioua et al, 2022) revealing characteristic lineage-specific consensus amino acid changes compared with the original 2019 Hu-1 sequence (Fig S1). As expected, the Omicron BA.1 strain is most divergent with a total of 33 aa changes in the S protein, whereas NL-02-2020 only differs by D614G from the original index virus. The impact of mutations in the S protein of VOCs has been the focus of many studies (Harvey et al, 2021; Jangra et al, 2021; Planas et al, 2021; Pastorio et al, 2022). The Delta VOC contains 22 amino acid changes outside of the spike, followed by 18 and 15 exchanges in Omicron and Alpha,

respectively. Currently, the functional impact of these alterations is poorly understood. The viral isolates used in this study show a few variations from the Covariants.org consensus sequence: Alpha (ORF1ab: N460Y), Gamma (ORF1ab: F3753V), Delta (ORF1ab: K261N, P2287S, P2046L; S: R683L; ORF3a D238Y), and Omicron BA.1 (ORF1ab: L6924F; S: E484K) (Fig S1).

Next, we analyzed the replication capacity of all SARS-CoV-2 variants in Calu-3 epithelial lung cells. Alpha and Gamma variants replicated with similar efficiency as the NL-02-2020 isolate in the absence of IFN, whereas replication of the Delta variant was ~2-fold increased (Fig 1A–C). In contrast, the Beta and Omicron variants produced 37 and 5,500-fold lower levels of viral RNA, respectively, than the early SARS-CoV-2 isolate (Fig 1A). To assess their ability to evade innate immune responses, we analyzed the impact of exogenous IFN types I, II, and III on the replication of SARS-CoV-2 VOCs. The treatment did not affect cell viability (Fig S2A) and all viruses replicated productively (Fig S2B). The spread of Omicron

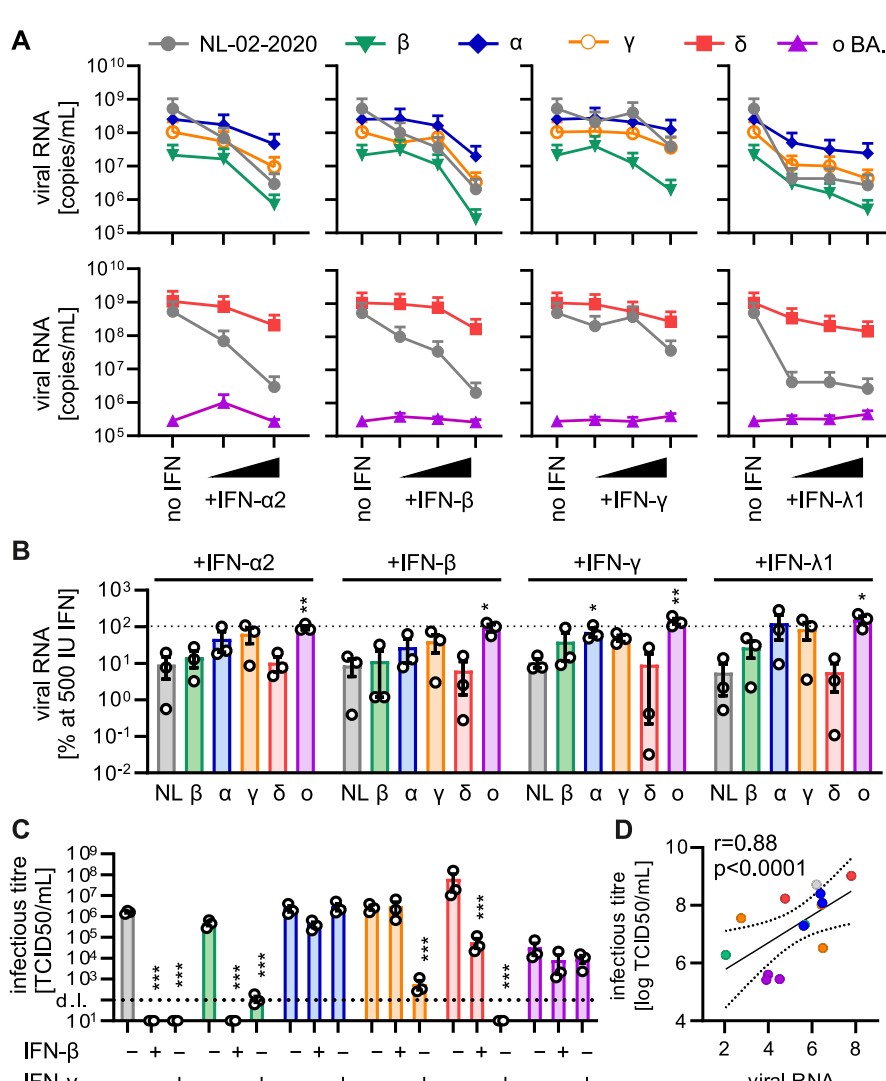

**Figure 1. IFN sensitivity of SARS-CoV-2 variants in Calu-3 cells.**

**(A)** Viral RNA in the supernatant of Calu-3 cells infected with indicated SARS-CoV-2 variants was quantified by qRT–PCR at 48 h postinfection (MOI 0.05). Cells were treated preinfection with increasing concentrations of indicated IFNs (α2 at 5 and 500 IU/ml, β, and γ at 5, 50, and 500 IU/ml or λ1 at 1, 10, and 100 ng/ml). Dots represent the mean of n = 3 + SEM. **(B)** Percentage of viral RNA in the supernatant compared with the no IFN control (set to 100%) at 500 IU/ml IFN (100 ng/ml for IFN-λ1) as indicated. Bars represent the mean of n = 3 ± SEM. **(C)** Infectious SARS-CoV-2 particles determined by the TCID50 assay in the supernatant of Calu-3 cells infected with indicated SARS-CoV-2 variants (MOI 0.05, 48 h postinfection). Cells were left untreated or were pretreated with 500 IU/ml IFN-β or IFN-γ. Bars represent the mean of n = 3 ± SEM. d.l., detection limit. **(B, C, D)** Correlation of infectious titer of viral particles ((C), TCID50) with viral RNA in the supernatant ((B), qRT–PCR), r, Pearson's correlation. Statistical significance was calculated using *t* tests. *P < 0.05; **P < 0.01; ***P < 0.001.

BA.1 was additionally confirmed using immunofluorescence assays for viral antigen expression showing single infected cells at 24 h postinfection and formation of large nucleocapsid- and spike-positive syncytia after 48 h (Fig S2C). Consistent with published results (Kim & Shin, 2021; Zanoni, 2021), the NL-02-2020 isolate was most sensitive towards IFN-β and IFN-λ1 (Fig 1A). All VOCs were still susceptible to exogenous IFNs, albeit not to the same extent as NL-02-20 (Fig 1A–C). In line with a recent report (Thorne et al, 2022), the Alpha variant was less restricted by all IFNs (Fig 1A and B). Beta and Delta were restricted by 500 IU/ml IFNs to a similar extent as NL-02-2020 (~10–20-fold less viral RNA yield in the supernatant), whereas Gamma showed an intermediate phenotype (Fig 1B). In contrast, replication of Omicron BA.1 was hardly affected by treatment with the four IFNs (Fig 1A and B). In line with the results on viral RNA production, infectious virus yields of Alpha and BA.1 were only marginally affected by IFN-β or IFN-γ, whereas all other variants showed drastic decreases in infectious virus production (Fig 1C). Of note, the Gamma variant was relatively resistant towards IFN-β treatment. The impact of IFNs on the different variants as analyzed by infectious virus yield correlated very well (r = 0.88, P < 0.0001) with the quantification of viral RNA in the supernatant (Fig 1D).

To determine the impact of exogenous IFN on SARS-CoV-2 VOCs in primary cell-derived settings, we used induced pluripotent stem cell (iPSC)-derived AT2 cells (iAT2) and air–liquid interface (ALI) cultures of fully differentiated primary human airway epithelial cells (HAEC). AT2 cells constitute ~60% of the pulmonary alveolar epithelial cells and are the main targets of SARS-CoV-2 in the distal lung (Delorey et al, 2021). Virus-induced loss of AT2 cells is linked to

the severity of COVID-19-associated acute respiratory distress syndrome (Gerard et al, 2021) and reduced lung regeneration (Delorey et al, 2021). On average, the Delta VOC produced ~7.0-fold higher and the Beta variant ~10-fold lower levels of viral RNA in infected iAT2 cells compared with the remaining SARS-CoV-2 isolates (Fig 2A). Of note, the Omicron VOC was only slightly attenuated compared to the NL-02-2020 strain in iAT2 cells in the absence of IFN (Fig 2A). In line with results obtained in Calu-3 cells, IFN-β and IFN-λ1 were also most effective in iAT2 cells and reduced RNA production by all SARS-CoV-2 variants by several orders of magnitude (Fig 2A and B). The metabolic activity of iAT2 cells was not affected by IFN treatment (Fig S2D). Only the Omicron variant was significantly less affected by IFN-β, IFN-γ, and IFN-λ1 in iAT2 cells (17, 2, and 29-fold reduction of RNA production, respectively) than NL-02-2020 (~100, 10, and 136-fold reduction, respectively; Fig 2B).

ALI cultures of primary HAEC contain a high proportion of the bronchial and tracheal ciliated epithelial cells, which are among the first target cell types during infection in vivo (Bridges et al, 2022). In the absence of exogenous IFNs, the Delta variant showed the highest viral RNA yields in all three donors. In contrast, the Omicron BA.1 variant displayed over 10,000-fold lower viral RNA yields than the Delta variant in two independent donors but replicated to similar levels as NL-02-2020 in a third donor (Fig S3A). However, the levels of Omicron BA.1 RNA in the supernatant generally increased over time indicating productive replication in all three donors (Fig S3B). Similar to Calu-3 cells, all IFNs reduced viral RNA of the SARS-CoV-2 variants in the apical mucus and intracellular in three independent donors. IFN-λ1 was the most potent inhibitor reducing

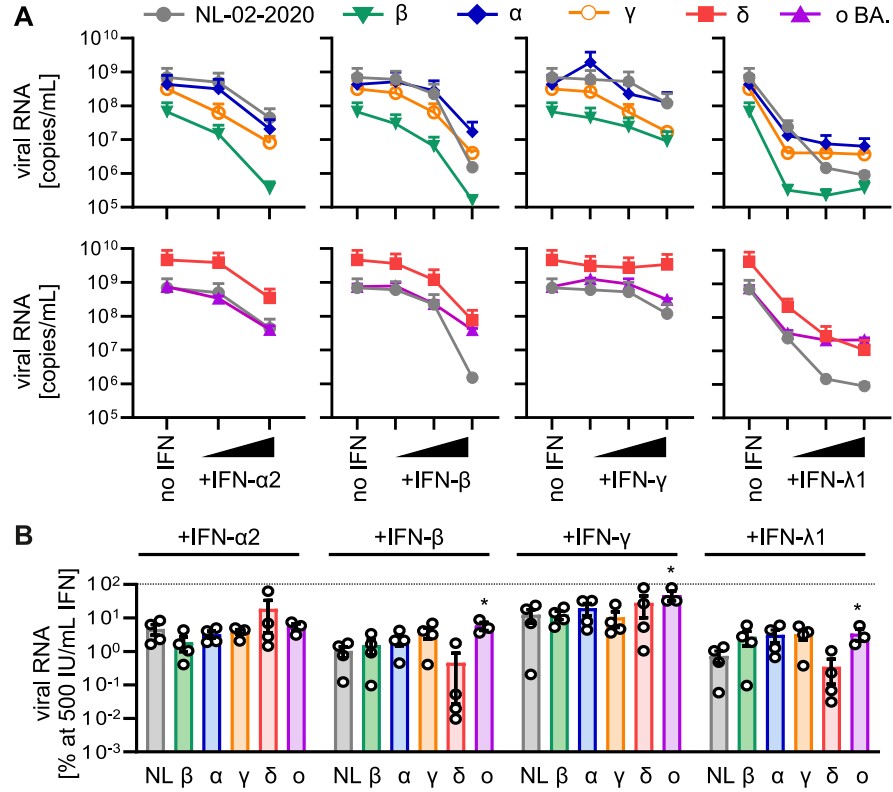

**Figure 2. IFN sensitivity of SARS-CoV-2 variants in iPSC-derived iAT2 cells.**
**(A)** Viral RNA in the supernatant of iAT2 cells infected with indicated SARS-CoV-2 variants was quantified by qRT–PCR at 48 h postinfection (MOI 0.5). The cells were treated 24 h preinfection with increasing concentrations of indicated IFNs (α2 at 5 and 500 IU/ml, β and γ at 5, 50, and 500 IU/ml or λ1 at 1, 10, and 100 ng/ml). Dots represent the mean of n = 3–4 + SEM. **(B)** Percentage of viral RNA in the supernatant compared with the no IFN control (set to 100%) at 500 IU/ml IFN (100 ng/ml for IFN-λ1) as indicated. Bars represent the mean of n = 3–4 ± SEM. Statistical significance was calculated using t tests. *P < 0.05.

viral RNA production by more than five orders of magnitude compared with the no IFN control at the highest concentration (Fig 3A). In line with the results obtained using Calu-3 and iAT2 cells, the Omicron BA.1 variant was least sensitive towards exogenous IFNs (Fig 3A). At highest concentrations of IFNs (500 IU/ml or 100 ng/ml), NL-02-2020 viral RNA yields were reduced >5,000-fold by IFN-$\alpha$2, $\beta$, and $\gamma$, and by more than six orders of magnitude by IFN-$\lambda$1 treatment. Overall, the Delta variant was ~100-fold less affected by IFN treatment than NL-02-2020. Again, the Omicron variant was almost fully resistant against all types of IFN. On average, its viral RNA yields were only reduced between 4- and 75-fold (Figs 3B and S3A).

All tested SARS-CoV-2 variants replicated productively in all three models: Calu-3 cells, iAT2 cells, and HAEC ALI cultures (Figs 4A, S2B and C, and S3B). Overall, the Delta variant yielded the highest RNA levels (~10-fold higher than NL-02-2020). The Beta variant produced about 20- to 50-fold less viral RNA than NL-02-2020 in iAT2 and Calu-3 cells. Replication of Omicron was strongly attenuated in Calu-3 cells (~500-fold) and two donors of ALI cultures (~500-fold), but comparable to NL-02-2020 in iAT2 cells or a third ALI culture donor (Fig 4A). Despite different growth rates of Omicron BA.1 in the three ALI culture donors, the impact of IFN was similar and generally modest. In line with this, replication capacity did not correlate (r = 0.12, P = 0.66) with restriction by IFNs (Fig 4B). In comparison, the inhibitory impact of IFNs on SARS-CoV-2 RNA yields

correlated (r = 0.52, P = 0.01) between Calu-3 and iAT2 cells (Fig 4C), indicating similar restrictive mechanisms in both cell types.

## Discussion

Our analyses show that Omicron BA.1 has reduced replication fitness compared to NL-02-2020 in all tested models (Fig 4A). However, BA.1 is highly resistant to the inhibitory effects of exogenous IFNs in all experimental settings. As recently reported (Pastorio et al, 2022), the infectivity of the Omicron BA.1 spike is lower than that of early 2020 SARS-CoV-2 isolates or the Delta VOC. Reduced susceptibility towards the IFN-induced antiviral state is, however, most likely conferred by mutations outside of the spike, in viral proteins that suppress or counteract innate immune defense mechanisms (Xia et al, 2020; Hayn et al, 2021). Further studies on the molecular determinants of reduced IFN sensitivity and improved innate immune evasion of emerging SARS-CoV-2 variants are highly warranted.

Omicron BA.1 is not the first variant that emerged with increased resistance to innate immune activation. Our results show that previous variants like Alpha are also more resistant against IFN treatment. This is in line with reports showing that Alpha expresses higher levels of known IFN antagonists such as ORF6

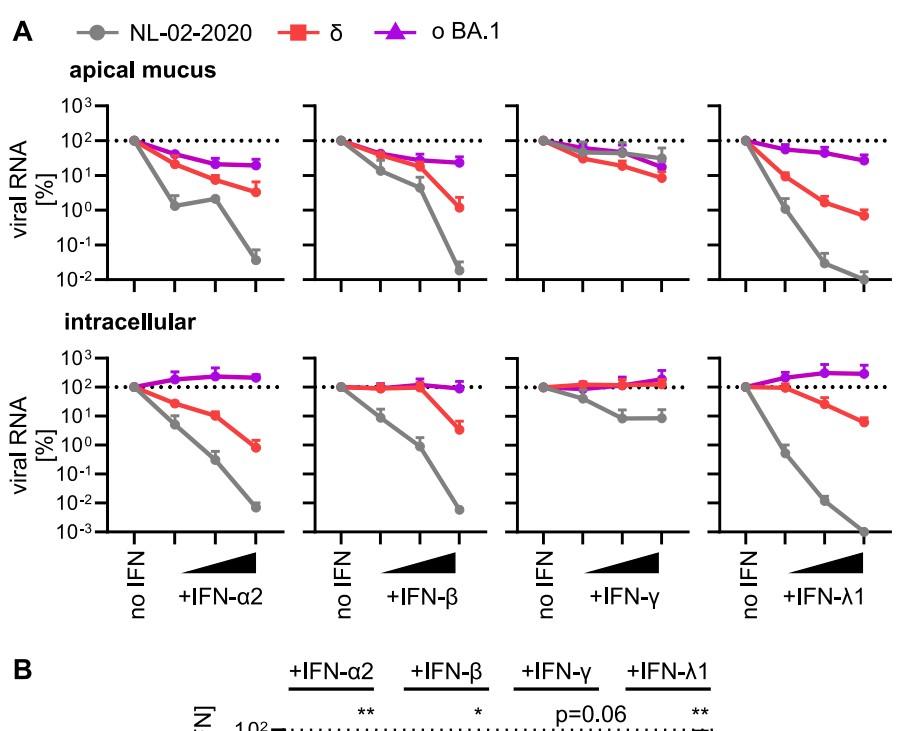

**Figure 3. IFN sensitivity of SARS-CoV-2 variants in ALI cultures.**
**(A)** Normalized viral RNA of ALI cultures from three donors (Donors A, B, and C) infected with indicated SARS-CoV-2 variants (MOI 0.5) quantified by qRT–PCR at 5 d postinfection. RNA was harvested from the apical surfaces (top panel) or intracellularly (bottom panel). ALI cultures were treated 1 h preinfection with increasing concentrations of indicated IFNs ($\alpha$2, $\beta$ and $\gamma$ at 5, 50, and 500 IU/ml or $\lambda$1 at 1, 10, and 100 ng/ml). All values are normalized to the no IFN control (set to 100%) and dots represent the mean of n = 3 ($\delta$ and o BA.1) or n = 2 (NL-02-2020) individual donors + SEM. **(B)** Percentage of viral RNA from the apical surfaces of ALI cultures compared with the no IFN control (set to 100%) at 500 IU/ml IFN (100 ng/ml for IFN-$\lambda$1) as indicated. Bars represent the mean of n = 3 ($\delta$ and o BA.1) or n = 2 (NL-02-2020), individual donors are represented by different symbols and connected with lines. Statistical significance was calculated using ratio paired t tests compared with NL-02-2020. *P < 0.05; **P < 0.01.

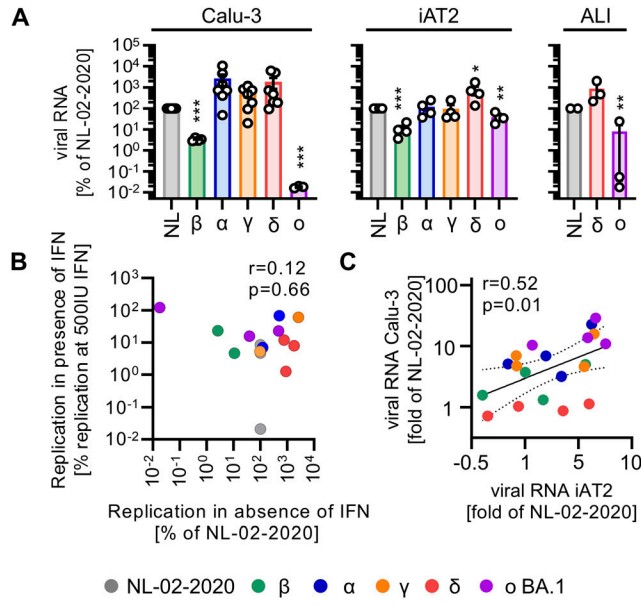

**Figure 4. Comparative analysis of SARS-CoV-2 variants in Calu-3 cells, iAT2 cells, and ALI cultures.**
**(A)** Normalized viral RNA in the supernatant of Calu-3 cells (left panel), iAT2 cells (middle panel) or ALI cultures (right panel, harvested from the apical surfaces) infected with indicated SARS-CoV-2 variants quantified by qRT–PCR at 2 d (Calu-3, iAT2) or 5 d (ALI cultures) postinfection (MOI 0.05, MOI 0.5, MOI 0.5 respectively). Bars represent the mean of n = 3–7 ± SEM or n = 3 individual donors (ALI cultures). **(A, B)** Scatter plot of viral RNA levels upon IFN treatment (average of all IFN treatments at 500 IU/ml or 100 ng/ml from Fig 1B) versus replication relative to NL-02-2020 (in Calu-3 cells from (A)) in the absence of IFN. **(C)** Scatter plot of viral RNA in the supernatant of Calu-3 or iAT2 cells as fold change compared with NL-02-2020 at maximum IFN concentrations (500 IU/ml or 100 ng/ml) (data from Figs 1B and 2B). Correlations in (B, C) (r and p) were calculated using Pearson's correlation. **(A)** Statistical significance was calculated using t tests ((A), left and middle panel) or ratio paired t tests ((A), right panel). *$P < 0.05$; **$P < 0.01$; ***$P < 0.001$.

(Thorne et al, 2022). The Gamma variant seems to be particularly resistant against IFN-$\beta$ but is still sensitive to other IFNs such as IFN-$\gamma$ in Calu-3 cells. Of note, the Delta variant is less susceptible to IFNs in primary HAEC ALI cultures but as sensitive as NL-02-2020 in Calu-3 or iAT2 cells. However, the Omicron BA.1 variant shows the most striking resistance against all three types of IFNs in all three models. In line with previous reports (Terenzi et al, 2007; Pervolaraki et al, 2018; Aso et al, 2019), this illustrates that IFNs induce distinct but overlapping antiviral programs in different cell types.

Previous studies have reported conflicting results on the resistance of Omicron towards IFNs. Initially, analyses of the growth of Omicron BA.1 in African green monkey Vero cells that are known to have defective IFN expression or human cell lines defective in innate immune sensors led to the speculation that Omicron BA.1 may be more sensitive to innate defenses than other virus strains (Bojkova et al, 2022a, 2022b). However, in agreement with our data, emerging evidence indicates that Omicron BA.1 is highly resistant against exogenous IFN (Guo et al, 2022; Shalamova et al, 2022).

Therapeutic induction of an antiviral state using IFNs remains a powerful tool against viral infections. However, current antiviral treatment with IFNs is associated with severe side effects (Saleki et al, 2021). Identification of the most effective IFN(s) is paramount to minimize adverse effects by reducing the dose required for

effective inhibition of SARS-CoV-2. Our results add to the accumulating evidence that type III IFN is particularly effective against SARS-CoV-2 (Felgenhauer et al, 2020; Stanifer et al, 2020; Vanderheiden et al, 2020; Hayn et al, 2021; Sposito et al, 2021 Preprint; Zanoni, 2021). In agreement with this, recent clinical studies showed that treatment of COVID-19 patients with IFN-$\lambda$ treatment promoted viral clearance and reduced hospitalization (Santer et al, 2022; Reis et al, 2023).

Expanding recent reports primarily using A549 or Calu-3 cell lines (Guo et al, 2022; Shalamova et al, 2022) our results in primary lung epithelial cell models for SARS-CoV-2 support the notion that humoral immune escape and effective IFN escape rather than increased replication capacity helped Omicron BA.1 to outcompete the Delta variant.

# Materials and Methods

## Cell culture

Calu-3 (human epithelial lung adenocarcinoma, kindly provided by Prof. Manfred Frick [Ulm University]) cells were cultured in Minimum Essential Medium Eagle (MEM, Cat#M4655; Sigma-Aldrich) supplemented with 10% (upon and after viral infection) or 20% (during all other times) heat-inactivated FBS (Cat#10270106; Gibco), 100 U/ml penicillin, 100 $\mu$g/ml streptomycin (Cat#15140122; Thermo Fisher Scientific), 1 mM sodium pyruvate (Cat#P04-8010; Pan Biotech), and 1x nonessential amino acids (Cat#M7145; Sigma-Aldrich). Vero E6 cells (*Cercopithecus aethiops*-derived epithelial kidney, ATCC Cat#CRL-1586) and TMPRSS2-expressing Vero E6 cells (kindly provided by the National Institute for Biological Standards and Control [NIBSC], No. 100978) were grown in DMEM (Cat#41965039; Gibco) supplemented with 2.5% (upon and after viral infection) or 10% (during all other times) heat-inactivated FBS (Cat#10270106; Gibco), 100 U/ml penicillin, 100 $\mu$g/ml streptomycin (Cat#15140122; Thermo Fisher Scientific), 2 mM L-glutamine (Cat#25030081; Gibco), 1 mM sodium pyruvate (Cat# P04-8010; Pan Biotech), 1x nonessential amino acids (Cat#M7145; Sigma-Aldrich), and 1 mg/ml Geneticin (Cat#10131-019; Gibco) (for TMPRSS2-expressing Vero E6 cells). Caco-2 cells (human epithelial colorectal adenocarcinoma, kindly provided by Prof. Holger Barth [Ulm University]) were grown in the same medium as Vero E6 cells but with supplementation of 10% heat-inactivated FBS.

Human-induced alveolar type 2 cells (iAT2) were differentiated from BU3 NKX2-1$^{GFP}$; SFTPC$^{tdTomato}$ induced pluripotent stem cells (Jacob et al, 2017) (iPCSs, kindly provided by Darrell Kotton, Boston University and Boston Medical Center) and maintained as alveolospheres embedded in 3D Matrigel in CK + DCI media, as previously described (Jacob et al, 2019). For infection studies, iAT2 cells were cultured as 2D cultures on Matrigel-coated plates in CK + DCI medium + 10 $\mu$M Y-27632 (Cat#1254; Tocris) for 48 h before switching to CK + DCI medium on day 3.

Primary HAEC were isolated from fresh lung tissue obtained during lung transplant or tumor resection with donor consent as previously described (Hoang et al, 2022; Lai et al, 2022) (ethics approval: Ethics Committee Medical School Hannover, Project number 2701-2015). Primary HAEC at passage 2 were thawed and expanded in a T75 flask in Airway Epithelial Cell Basal Medium (Cat#

C-21060; Promocell) supplemented with Airway Epithelial Cell Growth Medium Supplement Pack (Cat# C-39170; Promocell) and with 5 µg/ml Plasmocin Prophylactic (Cat# ant-mpp; InvivoGen), 100 µg/ml Primocin (Cat# ant-pm; InvivoGen), and 10 µg/ml Fungin (Cat# ant-fn; InvivoGen). The growth medium was replaced every two days until cells were 70–90% confluent, they were passaged using the DetachKIT (Cat#C-41200; Promocell) and plated at a density of 3.5 × 10$^4$ cells per 6.5 mm Transwell filter (Corning Costar 3470, Cat#3470; Corning Inc.) precoated with 30 µg/ml Collagen Solution (Cat#04902; StemCell Technologies). For differentiation, cells were plated with 200 µl growth medium apically and 600 µl basolaterally. The apical medium was replaced every 48 h till full confluency was reached and then removed completely to form an ALI. The basolateral medium was switched to differentiation medium and replaced every three days. Differentiation medium was prepared as previously described (Winkelmann et al, 2019; Lai et al, 2022). Cells were grown at ALI for 25 d, and from day 14 onwards the apical surface was incubated with 100 µl DPBS (Cat#14190144; Gibco) for 30 min every two days and then aspirated to remove accumulated mucus.

### SARS-CoV-2 stocks

The SARS-CoV-2 variant B.1.351 (Beta), 2102-cov-IM-r1-164 was provided by Prof. Michael Schindler (University of Tübingen) and the B.1.617.2 (Delta) variant by Prof. Florian Schmidt (University of Bonn). The BetaCoV/Netherlands/01/NL/2020 (NL-02-2020), B.1.1.7. (Alpha) and hCoV-19/Netherlands/NH-EMC-1720/2021, lineage B.1.1.529 (Omicron) variants were obtained from the European Virus Archive. The hCoV-19/Japan/TY7-503/2021 (Brazil P.1) (Gamma) (#NR-54982) isolate was obtained from BEI resources. SARS-CoV-2 strains were propagated on Vero E6 (NL-02-2020, Delta), VeroE6 overexpressing TMPRSS2 (Alpha), CaCo-2 (Beta) or Calu-3 (Gamma, Omicron) cells. To this end, 70–90% confluent cells in 75 cm$^2$ cell culture flasks were inoculated with the SARS-CoV-2 isolate (MOI of 0.03–0.1) in 3.5 ml serum-free medium (MEM, Cat#M4655; Sigma-Aldrich). The cells were incubated for 2 h at 37°C, before adding 20 ml medium containing 15 mM HEPES (Cat#6763.1; Carl Roth). Virus stocks were harvested as soon as a strong cytopathic effect (CPE) became apparent. The virus stocks were centrifuged for 5 min at 1,000$g$ to remove cellular debris, aliquoted, and stored at –80°C until further use.

### PFU assay

To determine the infectious titres, SARS-CoV-2 stocks were serially diluted 10-fold. Monolayers of Vero E6 cells in 12-wells were infected with the dilutions and incubated for 1–3 h at 37°C with shaking every 15–30 min. Afterwards, the cells were overlayed with 1.5 ml of 0.8% Avicel RC-581 (FMC Corporation) in the medium and incubated for 3 d. The cells were fixed by adding 1 ml 8% PFA (Cat#158127-100G; Sigma-Aldrich) in Dulbecco's phosphate buffered saline (DPBS, Cat#14190144; Gibco) and incubated at room temperature for 45 min. After discarding the supernatant, the cells were washed with DPBS (Cat#14190144; Gibco) once, and 0.5 ml of staining solution (0.5% crystal violet [Cat#42555; Carl Roth] and 0.1% triton X-100 [Cat#T8787; Sigma-Aldrich] in water) was added. After 20 min incubation at RT, the staining solution was removed using water,

virus-induced plaque formation was quantified, and PFU/ml were calculated.

### Effect of IFNs on SARS-CoV-2 replication

1.5 × 10$^5$ Calu-3 cells were seeded in 24-well plates. 24 and 96 h post-seeding, the cells were stimulated with increasing amounts of IFNs ($\alpha$2 [Cat#11101-2; R&D Systems], $\beta$ [Cat#8499-IF-010/CF; R&D Systems], and $\gamma$ [Cat#285-IF-100/CF; R&D Systems] using 5, 50, and 500 IU/ml or IFN-$\lambda$1 [Cat#1598-IL-025/CF; R&D Systems] using 1, 10, and 100 ng/ml) in 0.5 ml of medium. 6–12 h after the first stimulation, the medium was replaced. 2 h after the second stimulation, Calu-3 cells were infected with the indicated SARS-CoV-2 variants (MOI 0.05) and 5 h later, the cells were washed once with DPBS (Cat#14190144; Gibco) and 0.5 ml fresh medium was added. At 6 h (wash control) and 48 h postinfection, supernatants were harvested for qRT–PCR and Tissue culture infection dose 50 (TCID50) analysis.

1.5 × 10$^4$–3 × 10$^4$ iAT2 cells were seeded as single cells in 96-well plates coated for 1 h at 37°C with 0.16 mg/ml Matrigel (Cat#356238; Corning) diluted in DMEM/F12 (Cat#11330032; Thermo Fisher Scientific). 48 h post seeding, the cells were stimulated with increasing amounts of IFNs, IFN$\alpha$2 (Cat#11101-2; R&D Systems), IFN$\beta$ (Cat#8499-IF-010/CF; R&D Systems), and IFN$\gamma$ (Cat#285-IF-100/CF; R&D Systems) using 5, 50, and 500 IU/ml or IFN$\lambda$1 (Cat#1598-IL-025/CF; R&D Systems) using 1, 10, and 100 ng/ml, in 150 µl medium. 24 h posttreatment, iAT2 cells were infected with the indicated SARS-CoV-2 variants (MOI 0.5). 5–6 h later, supernatants were removed, cells were washed once with DPBS (Cat#14190144; Gibco), and 200 µl of fresh medium was added. Supernatants were harvested at 6 h (wash control) and 48 h postinfection for qRT–PCR analysis.

ALI cultures were grown until experiments were performed. Immediately before treatment and infection, the apical surface of the ALI cultures was washed three times with 200 µl 37°C DPBS (Cat#14190144; Gibco) to remove accumulated mucus. Then, the cells were stimulated with increasing amounts of IFNs ($\alpha$2 [Cat#11101-2; R&D Systems], $\beta$ [Cat#8499-IF-010/CF; R&D Systems], and $\gamma$ [Cat#285-IF-100/CF; R&D Systems] using 5, 50, and 500 IU/ml or IFN$\lambda$1 [Cat#1598-IL-025/CF; R&D Systems] using 1, 10, and 100 ng/ml) added to 700 µl of the basal medium. 1 h later, the cells were infected with 0.5 MOI of indicated SARS-CoV-2 VoCs added to the apical surface. After incubation for 2 h at 37°C, the viral inoculum was removed, and the cells were washed three times with 200 µl 37°C DPBS (Cat#14190144; Gibco) and again cultured as ALI. The DPBS (Cat#14190144; Gibco) from the third washing step was kept as day 0 control for qRT–PCR. At 3- and 5-d postinfection, shed viral particles were harvested by incubating the apical surface with 150 µl, 37°C DPBS (Cat#14190144; Gibco) for 15 min at RT. The PBS containing the mucus and the viral particles and the cells were used in qRT–PCR analysis.

### qRT–PCR

N (nucleoprotein) transcript levels were determined in supernatants collected from SARS-CoV-2-infected Calu-3 or iAT2 cells at 48 h postinfection, as previously described (Nchioua et al, 2020). N transcript levels were also determined in the PBS harvested from the apical surfaces of ALI cultures at 3- and 5-d postinfection and in the cells at 5-d postinfection. Total RNA was isolated using the Viral

RNA Mini Kit (Cat#52906; QIAGEN) according to the manufacturer's instructions. qRT–PCR was performed as previously described (Nchioua et al, 2020) using TaqMan Fast Virus 1-Step Master Mix (Cat#4444436; Thermo Fisher Scientific) and a OneStepPlus Real-Time PCR System (96-well format, fast mode). Primers for N were purchased from Biomers and dissolved in RNase-free water (Cat#10977015; Invitrogen). Synthetic SARS-CoV-2-RNA (Cat#102024; Twist Bioscience) or RNA isolated from BetaCoV/France/IDF0372/2020 viral stocks quantified via this synthetic RNA (for low Ct samples) was used as a quantitative standard to obtain viral copy numbers. All reactions were run in technical duplicates. Forward primer (HKU-NF): 5′-TAATCAGACAAGGAACTGATTA-3′; Reverse primer (HKU-NR): 5′-CGAAGGTGTGACTTCCATG-3′; Probe (HKU-NP): 5′-FAM-GCAAATTGTG-CAATTTGCGG-TAMRA-3′. GAPDH primer/probe sets (Cat# 4310884E; Thermo Fisher Scientific) were used for normalization of cellular RNA levels for ALI cultures.

### TCID50 endpoint titration

SARS-CoV-2 stocks or infectious supernatants were serially diluted. $1.5 \times 10^4$ Caco-2 cells were seeded per well in 96-well F-bottom plates in 100 $\mu$l medium and incubated overnight. Next, 100 $\mu$l of diluted SARS-CoV-2 stocks or infectious supernatants were used for infection, resulting in final dilutions of 1:10$^1$ to 1:10$^{12}$ on the cells in eight technical replicates. The cells were then incubated for at least 5 d and monitored for CPE. TCID50/ml was calculated according to the Reed–Muench method.

### MTT (3-[4,5-dimethyl-2-thiazolyl]-2,5-diphenyl-2H-tetrazolium bromide) assay

$6 \times 10^4$ Calu-3 cells were seeded in 96-well F-bottom plates. $1.5 \times 10^4$–$3 \times 10^4$ iAT2 cells were seeded as single cells in 96-well F-bottom plates, coated for 1 h at 37°C with 0.16 mg/ml Matrigel (Cat#356238; Corning), and diluted in DMEM/F12 (Cat#11330032; Thermo Fisher Scientific). The cells were stimulated with increasing amounts of IFNs ($\alpha$2 [Cat#11101-2; R&D Systems], $\beta$ [Cat#8499-IF-010/CF; R&D Systems], and $\gamma$ [Cat#285-IF-100/CF; R&D Systems] using 5, 50, and 500 IU/ml or IFN-$\lambda$1 [Cat#1598-IL-025/CF; R&D Systems] using 1, 10, and 100 ng/ml) in 100 $\mu$l medium 24 and 96 h post-seeding. 6 h after the first stimulation, the medium was replaced. To analyze the cell viability of Calu-3 cells and iAT2 cells after IFN treatment, 100 $\mu$l of MTT solution (0.5 mg/ml in DPBS [Cat#14190144; Gibco]) was added to the cells 2 h after the second stimulation and the cells were incubated for 3 h at 37°C. Subsequently, the supernatant was discarded and DPBS (Cat#14190144; Gibco) was added for 20 min. After washing with 100 $\mu$l DPBS (Cat#14190144; Gibco), the formazan crystals were dissolved in 100 $\mu$l of a 1:2 mixture of DMSO (Cat#D12345; Invitrogen) and ethanol. Absorption was measured at 490 nm with the baseline corrected at 650 nm by using a Vmax kinetic microplate reader (Molecular Devices) and the SoftMax Pro 7.0.3 software.

### Immunofluorescence

$1.5 \times 10^5$ Calu-3 cells were seeded on glass cover slips in 24-well plates. 36 h after seeding, the cells were washed with DPBS (Cat#14190144; Gibco) and infected with 0.05 MOI of the omicron BA.1 variant. 24 and 48 h after the infection, the cells were washed with DPBS (Cat#14190144; Gibco) and fixed for 30 min in 4% PFA in PBS (Cat#sc-281692; Santa Cruz Biotechnology). Next, the cells were washed three times with cold DPBS (Cat#14190144; Gibco) and blocked and permeabilized with DPBS (Cat#14190144; Gibco) containing 0.5% Triton X-100 (Cat#T8787; Sigma-Aldrich) and 5% FBS (Cat#10270106; Gibco) for 30 min at RT. The cells were then washed twice with cold DPBS (Cat#14190144; Gibco) and incubated with primary antibodies diluted in DPBS (Cat#14190144; Gibco) (Anti-SARS spike glycoprotein antibody, 1:300; Abcam, Cat#ab273433, SARS-CoV/SARS-CoV-2 Nucleocapsid Antibody, 1:300, Cat#40143-R001; Sino Biological) for 2 h at 4°C. The cells were washed three times with DPBS (Cat#14190144; Gibco) containing 0.1% Tween 20 (Cat# P9416; Sigma-Aldrich) and incubated with DAPI (1 mg/l, Cat#D1306; Invitrogen) and secondary antibodies diluted in DPBS (Cat#14190144; Gibco) (goat anti-mouse IgG (H + L), 1:400; Invitrogen, A-11001, goat anti-rabbit IgG (H + L), 1:400; Invitrogen, A-11011) for 2 h at 4°C. The cells were then washed three times with DPBS (Cat#14190144; Gibco) containing 0.1% Tween 20 (Cat# P9416; Sigma-Aldrich) and once with deionized water. The cover slips were then mounted on microcopy slides using self-hardening Mowiol mounting medium and dried at 4°C overnight (Koepke et al, 2020). The samples were analyzed using a Zeiss LSM 710 confocal microscope. The images were processed using the ImageJ software.

### Statistical analysis

Statistical analysis was performed using GraphPad Prism software. Two-tailed unpaired $t$ tests or ratio paired $t$ tests (for donor experiments) were used to determine statistical significance. Significant differences are indicated as: *$P$ < 0.05; **$P$ < 0.01; ***$P$ < 0.001 or as the $P$-value above the respective bars. Nonsignificant is not indicated. Statistical parameters are further specified in the figure legends.

# Supplementary Information

# Acknowledgements

We thank Daniela Krnavek, Martha Mayer, Kerstin Regensburger, Regina Burger, Jana-Romana Fischer, and Birgit Ott for assistance. We also thank Michael Schindler (Tübingen University) for providing the B.1.351 (Beta) variant and Florian Schmidt (Bonn University), Beate M. Kümmerer (Bonn University), and Hendrick Streeck (Bonn University) for providing the B.1.617.2 (Delta) variant. The following reagent was obtained through BEI Resources, NIAID, NIH: SARS-Related Coronavirus 2, Isolate hCoV-19/Japan/TY7-503/2021 (Brazil P.1), NR-54982, contributed by the National Institute of Infectious Diseases. hCoV-19/Netherlands/01/NL/2020 and nCoV19 isolate/England/MIG457/2020 were provided via the European Virus Archive. The BU3-NGST iPSC line was generated with funding from the National Center for Advancing Translational Sciences ("NCATS") (grant U01TR001810) and kindly provided by DN Kotton (Center for Regenerative Medicine, Boston Medical Center). This study was supported by DFG grants to F Kirchhoff, KMJ Sparrer, D Kmiec, and M Frick (F Kirchhoff and KMJ Sparrer, CRC 1279 and SPP1923; M Frick, 278012962, and

458685747), the BMBF to F Kirchhoff and KMJ Sparrer (Restrict SARS-CoV-2 and IMMUNOMOD-01KI2014) and a COVID-19 research grant from the Ministry of Science, Research and the Arts of Baden-Württemberg (MWK) to F Kirchhoff. KMJ Sparrer was additionally supported by a COVID-19 emergency grant from the DFG (SP1600/6-1). D Kmiec received funding from the European Union's Horizon 2020 research and innovation programme under the Marie Sklodowska-Curie grant agreement No. 101062524. A joint project (Bay-VOC) from the Bavarian State Ministry of the Environment and Public Health supported A Graf, S Klute, and H Blum. L Koepke, S Klute, M Hirschenberger, S Noettger, and A Schundner are part of the International Graduate School for Molecular Medicine, Ulm (IGradU). D Kmiec is supported by the Baustein Grant from Ulm University.

## Author Contributions

R Nchioua: conceptualization, data curation, formal analysis, investigation, methodology, and writing—review and editing.

A Schundner: resources, investigation, methodology, and writing—review and editing.

S Klute: data curation, formal analysis, investigation, visualization, and writing—review and editing.

L Koepke: data curation, formal analysis, investigation, visualization, and writing—review and editing.

M Hirschenberger: investigation and writing—review and editing.

S Noettger: formal analysis, investigation, and writing—review and editing.

G Fois: resources, investigation, and writing—review and editing.

F Zech: investigation, methodology, and writing—review and editing.

A Graf: formal analysis, investigation, and writing—review and editing.

S Krebs: resources, formal analysis, investigation, methodology, and writing—review and editing.

P Braubach: resources.

H Blum: resources, supervision, funding acquisition, and writing—review and editing.

S Stenger: resources and writing—review and editing.

D Kmiec: formal analysis, supervision, methodology, and writing—review and editing.

M Frick: resources, supervision, funding acquisition, and writing—review and editing.

F Kirchhoff: resources, formal analysis, supervision, funding acquisition, visualization, project administration, and writing—original draft, review, and editing.

KMJ Sparrer: conceptualization, resources, data curation, supervision, funding acquisition, investigation, visualization, project administration, and writing—original draft, review, and editing.

## Conflict of Interest Statement

The authors declare that they have no conflict of interest.

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
