## [Reviewer comments · Life Science Alliance]

Life Science Alliance

Reduced replication but increased interferon resistance of SARS-CoV-2 Omicron BA.1

Rayhane Nchioua, Annika Schundner, Susanne Klute, Lennart Koepke, Maximilian Hirschenberger, Sabrina Noettger, Giorgio Fois, Fabian Zech, Alexander Graf, Stefan Krebs, Peter Braubach, Helmut Blum, Steffen Stenger, Dorota Kmiec, Manfred Frick, Frank Kirchhoff, and Konstantin Sparrer

DOI: <https://doi.org/10.26508/lsa.202201745>

Corresponding author(s): Konstantin Sparrer, University Hospital Ulm and Frank Kirchhoff, University of Ulm

Review Timeline:

Submission Date:	2022-09-29
Editorial Decision:	2022-11-25
Revision Received:	2023-02-16
Editorial Decision:	2023-03-14
Revision Received:	2023-03-18
Accepted:	2023-03-20

Scientific Editor: Novella Guidi

Transaction Report:

November 25, 2022

Re: Life Science Alliance manuscript #LSA-2022-01745-T

Dr. Konstantin M.J. Sparrer
University Hospital Ulm
Meyerhofstr.1
Ulm, BW 89081
Germany

Dear Dr. Sparrer,

Thank you for submitting your manuscript entitled "Reduced replication but increased interferon resistance of SARS-CoV-2 Omicron BA.1" to Life Science Alliance. The manuscript was assessed by expert reviewers, whose comments are appended to this letter. We invite you to submit a revised manuscript addressing the Reviewer comments.

Thank you for this interesting contribution to Life Science Alliance. We are looking forward to receiving your revised manuscript.

Sincerely,

B. MANUSCRIPT ORGANIZATION AND FORMATTING:

Reviewer #1 (Comments to the Authors (Required)):

In their paper, Nchioua et al. investigate the influence of interferons on the replication of SARS-CoV-2 variants. The study is very carefully conducted and compares the effect of different interferons on the viral replication of SARS-CoV-2 VOCs. The effect of interferons is compared in the cell line Calu3, with iPSC-generated iAT2 cells, and in ALI cultures. Detection of the virus is performed by RT-PCR with the exception of TCID50 in Calu3 cells after IFN-gamma treatment.

Major comments

- Detection of viral replication is almost exclusively by RT-PCR. Only for IFN-gamma treatment in Calu3 cells was a TCID50 assay performed. It is unclear why IFN- γ was chosen for this assay, as it had the least antiviral effect. A comparison with a TCID50 value after IFN-beta treatment would confirm the statement.
- For Omicron, no effect of IFNs on viral replication was shown. However, a very low replication rate of the virus was shown in cell culture. It is questionable whether the virus can replicate in these cells. To show this, an alternative methodological approach, e.g. detection of virus infection and spread in the course of the infection by immunohistochemical staining, would be desirable.
- In Figure 3, the difference between Omicron and other VOCs in ALI cultures is shown. On the one hand, it is unclear why donor A was not tested for strain NL-02-2020. On the other hand, it is also not clear whether the Omicron virus can replicate at all in an ALI culture. Immunohistological detection of SARS-CoV-2 proteins should be shown.
- A differentiation to other studies investigating similar questions is missing. Here, the novelty of the data in relation to other studies investigating similar mechanisms have to be shown. Difference and similarities have to be mentioned and possible interpretations should be pointed out.

Minor comments

- Material and Methods describes that the values after infection were subtracted from those after 48h. It should be shown in the text or as a figure how high the increase of viral RNA is in comparison. Especially when showing the omicron variant, this would show if there is a productive infection at all

Reviewer #2 (Comments to the Authors (Required)):

Here the authors have investigated the replication fitness and the sensitivity to type I, II and III IFN of the major SARS-CoV-2 VOCs on three different cell models, Calu-3 cell-line, iAT2 and primary epithelial ALI cultures.

The data suggest that replication of Omicron variant BA.1 is highly attenuated in Calu-3 and primary ALI culture but not in iAT2 cultures. Beta variant also showed less replication fitness in Calu-3 and iAT2 cells. The sensitivity to different IFN was investigated and the data suggest that the Omicron variant is resistant to the different IFNs used, whereas variation in IFN sensitivity is observed with the other variants. Alpha variant seems similarly resistant to IFNs in Calu-3 cells same as observed for Delta. The authors discuss that the resistance to innate immunity by Omicron and not replication capacity might contribute to spread of the VOC.

The results are clearly described and it is clear that there are differences between the VOCs regarding fitness and IFN sensitivity. However, the importance of these differences remain unclear as most differences seem to depend on the cell model used and it is unclear why these different models have been chosen. It would have been more appropriate to choose the best model ie primary epithelial ALI model. Why are the results different between modes, could this be due to different IFN responses? Restriction to IFN does not seem to be the only the case for Omicron but also some other VOCs and this needs to be discussed.

Specific concerns

- The primary epithelial cell ALI cultures seem to be most representative to the airways infections but here only two donors have been used and not all VOCs have been tested which makes comparison more difficult.
- replication efficiency is very different for the VOCs and this might also somehow impact the IFN sensitivity. Low viral replication of Omicron correlates with less restriction by IFN. It would be interesting to compare IFN sensitivity when viral RNAs/infection are similar even if this would require different MOIs, especially for the Calu-3 experiments where Omicron replication is quite some logs lower.

- how effective are the different IFNs in activating antiviral programs. The induction of IFN-stimulated genes need to be shown to understand the differences observed between cell-lines as well as the efficacy of the different IFNs to activate antiviral immunity.
- it might be that the difference in viral replication and IFN sensitivity observed between VOCs might be due to intrinsic activation of type I IFN responses in the cells and it might be interesting to investigate the ISG induction in these cells upon infection by the VOCs.

Point by Point response:

We thank both reviewers for their positive feedback (Reviewer 1 “study is very carefully conducted” and Reviewer 2 “The results are clearly described and it is clear that there are differences between the VOCs regarding fitness and IFN sensitivity”) on our work and their constructive suggestions, which strengthened our manuscript.

Reviewer #1 (Comments to the Authors (Required)):

In their paper, Nchioua et al. investigate the influence of interferons on the replication of SARS-CoV-2 variants. The study is very carefully conducted and compares the effect of different interferons on the viral replication of SARS-CoV-2 VOCs. The effect of interferons is compared in the cell line Calu3, with iPSC-generated iAT2 cells, and in ALI cultures. Detection of the virus is performed by RT-PCR with the exception of TCID50 in Calu3 cells after IFN-gamma treatment.

Major comments

- Detection of viral replication is almost exclusively by RT-PCR. Only for IFN-gamma treatment in Calu3 cells was a TCID50 assay performed. It is unclear why IFN- γ was chosen for this assay, as it had the least antiviral effect. A comparison with a TCID50 value after IFN-beta treatment would confirm the statement.

IFN- γ was chosen as an example to show that small differences in the qPCR may translate to significant impact in infectious virus yield (Fig. 1 A and C). To address the reviewer’s point, we performed TCID50 assays of the IFN- β treated samples (updated Fig. 1C) and show that the infectious virus yields correlate very well ($r=0.88$) with the viral RNA levels (new Fig. 1 D). This agrees with our previous data showing that viral loads in culture supernatants correlate well with TCID50 or plaque assay results (Hayn et al, Cell Reports, 2021, Fig. 6D-F ; Prelli Bozzo et al, Nat Comm, 2021, Fig. 1H).

- For Omicron, no effect of IFNs on viral replication was shown. However, a very low replication rate of the virus was shown in cell culture. It is questionable whether the virus can replicate in these cells. To show this, an alternative methodological approach, e.g. detection of virus infection and spread in the course of the infection by immunohistochemical staining, would be desirable.

Despite low replication rates of Omicron BA.1 in various tissues (Hui et al, eBio Medicine, 2022; Nchioua et al, Signal transduction and targeted therapy, 2022) infectious virus production was readily detected in the supernatant of Calu-3 cells (updated Figure 1C). To further demonstrate that Omicron spreads in Calu-3 cells, we used immunofluorescence analyses (new Figure S2C). Whereas after 24h only individual cells were infected (Spike and nucleocapsid positive), syncytia formation was observed at 48h post infection (new Figure S2C) indicating active replication and viral spread.

- In Figure 3, the difference between Omicron and other VOCs in ALI cultures is shown. On the one hand, it is unclear why donor A was not tested for strain NL-02-2020. On the other hand, it is also not clear whether the Omicron virus can replicate at all in an ALI culture. Immunohistological detection of SARS-CoV-2 proteins should be shown.

For Donor A, the NL-02-2020 sample was unfortunately technically lost. We have added an additional donor (Donor C), which confirms the results obtained using the two previous donors.

ALI cultures are a well-established and relevant model to study SARS-CoV-2 infections. Analysis of the viral RNA loads 6 h, 3 days and 5 days post infection showed, that despite overall lower levels in the first two donors, Omicron RNA levels increased, indicating ongoing replication (new Figure S3B).

Analysis of BA.1 in the new Donor C showed that despite similar replication levels as NL-02-2020, it is still more resistant to IFNs than NL-02-2020 (new Figure 3A and B, new Figure S3A).

- A differentiation to other studies investigating similar questions is missing. Here, the novelty of the data in relation to other studies investigating similar mechanisms have to be shown. Difference and similarities have to be mentioned and possible interpretations should be pointed out.

In line with our data, the alpha variant was previously shown to be more resistant against innate immune activation (Thorne et al, Nature, 2021). Previous studies have reported conflicting results, suggesting that Omicron BA.1 is more sensitive to innate host responses (Bojkova et al, Cell Research, 2022; Bojkova et al, Cell Discovery, 2022) or less sensitive to IFN (Guo et al, PNAS, 2022; Shalamova et al, PNAS Nexus, 2022). We discuss this now in detail (lines 173-179).

Minor comments

- Material and Methods describes that the values after infection were subtracted from those after 48h. It should be shown in the text or as a figure how high the increase of viral RNA is in comparison. Especially when showing the omicron variant, this would show if there is a productive infection at all

We thank the reviewer for mentioning this important issue. All data are now displayed as raw values and the figure legends have been updated. For Calu-3 cells, we now show the wash controls (6 h post infection) in comparison to the 48h non IFN treated samples (new Fig S2B and E). Viral RNA levels for the omicron variant were not detectable in the wash control, but increased to 10^5 at 48h in Calu-3 cells. For the ALI cultures, we now also show the wash controls (2 h post infection) in comparison to the viral RNA levels at 3 days and 5 days post infection (new Fig S3B). In general, the Omicron BA.1 RNA levels at 3 days and 5 days post infection were more than two logs above those obtained the 2h control for Donor A and B and >5 logs above background for Donor C. In all cases this indicates productive infection of Omicron BA.1.

Reviewer #2 (Comments to the Authors (Required)):

Here the authors have investigated the replication fitness and the sensitivity to type I, II and III IFN of the major SARS-CoV-2 VOCs on three different cell models, Calu-3 cell-line, iAT2 and primary epithelial ALI cultures.

The data suggest that replication of Omicron variant BA.1 is highly attenuated in Calu-3 and primary ALI culture but not in iAT2 cultures. Beta variant also showed less replication fitness in Calu-3 and iAT2 cells. The sensitivity to different IFN was investigated and the data suggest that the Omicron variant is resistant to the different IFNs used, whereas variation in IFN sensitivity is observed with the other variants. Alpha variant seems similarly resistant to IFNs in Calu-3 cells same as observed for Delta. The authors discuss that the resistance to innate immunity by Omicron and not replication capacity might contribute to spread of the VOC.

The results are clearly described and it is clear that there are differences between the VOCs regarding fitness and IFN sensitivity. However, the importance of these differences remain unclear as most differences seem to depend on the cell model used and it is unclear why these different models have been chosen. It would have been more appropriate to choose the best model ie primary epithelial ALI model. Why are the results different between models, could this be due to different IFN responses? Restriction to IFN does not seem to be the only the case for Omicron but also some other VOCs and this needs to be discussed.

The reviewer raises a very important point. The strength of the IFN responses i.e. the levels and upregulated ISGs are dependent on the cell model used. We strongly believe that the most relevant model are the ALI cultures as mentioned in our manuscript. To further strengthen the ALI culture data, we added an independent third donor. Notably, however, Omicron BA.1 was the least IFN-sensitive variant in all models used. We now discuss that other variants are also less restricted by IFN than early SARS-CoV-2 and that there are cell type dependent differences (lines 163-172).

Specific concerns

- The primary epithelial cell ALI cultures seem to be most representative to the airways infections but here only two donors have been used and not all VOCs have been tested which makes comparison more difficult.

A third independent donor (Donor C) has been added. It supports the conclusions from Donor A and B that Omicron BA.1 is least sensitive to IFNs (new Fig. 3A and new Fig 3B). However, unlike for the first two donors, Omicron BA.1 replicated more efficiently than NL-02-2020 in Donor C. Due to the limited number of ALI cultures that can be obtained from a single donor and long differentiation time (over a month) we focused on Delta and Omicron in comparison to NL-02-2020.

- replication efficiency is very different for the VOCs and this might also somehow impact the IFN sensitivity. Low viral replication of Omicron correlates with less restriction by IFN. It would be interesting to compare IFN sensitivity when viral RNAs/infection are similar even if this would require different MOIs, especially for the Calu-3 experiments where Omicron replication is quite some logs lower.

In the third ALI culture donor, the BA.1 RNA levels in the supernatant were similar to those of NL-02-2022 (new Fig. 3A and B, updated Fig. 4A). Of note, in iAT2 cells the replication capacity of BA.1 was also similar to NL-02-2020 and it was still resistant towards exogenous IFN. Correlation analyses showed that IFN sensitivity did not correlate with replication efficiency in iAT2 and Calu-3 cells (updated Fig. 4B).

- how effective are the different IFNs in activating antiviral programs. The induction of IFN-stimulated genes need to be shown to understand the differences observed between cell-lines as well as the efficacy of the different IFNs to activate antiviral immunity.

The reviewer raises a relevant point. Depending on the cell type, different IFNs are known to induce divergent but overlapping transcriptional programs by upregulating >600 ISGs (Schoggins, Annu Rev Virol, 2019). However, the anti-viral impact of IFNs is not dependent on the average levels of ISG induction, but determined by the levels of a limited number of specific ISGs that are capable of restricting the virus. We have previously shown that ZAP contributes to the antiviral effect of IFNs (Nchioua et al., mBio 2020). In contrast, strongly IFN-upregulated IFITM1-3 promote rather than restrict SARS-CoV-2 infection (Prelli Bozzo et al, Nat Comm, 2021). Thus, higher ISG induction is not necessarily synonymous with stronger virus inhibition. The identification of further IFN-inducible factors that restrict SARS-CoV-2 is of interest but beyond the scope of this study.

- it might be that the difference in viral replication and IFN sensitivity observed between VOCs might be due to intrinsic activation of type I IFN responses in the cells and it might be interesting to investigate the ISG induction in these cells upon infection by the VOCs.

Intrinsic activation of the IFN system by the WT virus isolates used in this study is expected to be very low, as they evolved diverse mechanisms to prevent innate immune detection/signaling (Lee et al, MPMI, 2022; Minkoff et al, Nature Rev Microbiol, 2023). We found that OAS1 and ISG15 (as highly

IFN inducible genes) are induced by virus infection alone but at low levels in ALI cultures (~4.9-fold for OAS1 and 3.2-fold for ISG15) that did not differ significantly between NL-02-2020, Delta and Omicron BA.1.

March 14, 2023

RE: Life Science Alliance Manuscript #LSA-2022-01745-TR

Dr. Konstantin M.J. Sparrer
University Hospital Ulm
Meyerhofstr.1
Ulm, BW 89081
Germany

Dear Dr. Sparrer,

Thank you for submitting your revised manuscript entitled "Reduced replication but increased interferon resistance of SARS-CoV-2 Omicron BA.1". We would be happy to publish your paper in Life Science Alliance pending final revisions necessary to meet our formatting guidelines.

- please address the final Reviewer 2's comment
- please add ORCID ID for secondary corresponding author-you should have received instructions on how to do so
- please add a category for your manuscript to our system
- please add the Twitter handle of your host institute/organization as well as your own or/and one of the authors in our system
- please double-check your figure callouts for Figure S2; you have a callout for Figure S2E, but this is not in the legend or the figure, and you are missing a callout for Figure S2D

A. FINAL FILES:

B. MANUSCRIPT ORGANIZATION AND FORMATTING:

Sincerely,

Reviewer #1 (Comments to the Authors (Required)):

The authors answered all open questions

Reviewer #2 (Comments to the Authors (Required)):

The authors have addressed most of my comments. One comment remains that is not really addressed.

Original comment: how effective are the different IFNs in activating antiviral programs. The induction of IFN-stimulated genes need to be shown to understand the differences observed between cell-lines as well as the efficacy of the different IFNs to activate antiviral immunity.

The authors have misunderstood the question as it is indeed beyond the scope to identify genes that restrict SARS-CoV-2. However, it would be helpful to show that ISGs are induced by the different IFNs to assess whether the IFNs act in the same way and to explain the observed differences.

Point-by-point response.

The authors have addressed most of my comments. One comment remains that is not really addressed.

Original comment: how effective are the different IFNs in activating antiviral programs. The induction of IFN-stimulated genes need to be shown to understand the differences observed between cell-lines as well as the efficacy of the different IFNs to activate antiviral immunity.

The authors have misunderstood the question as it is indeed beyond the scope to identify genes that restrict SARS-CoV-2. However, it would be helpful to show that ISGs are induced by the different IFNs to assess whether the IFNs act in the same way and to explain the observed differences.

To address the reviewers remaining comment, we analysed the induction of a set of ISGs (out of ~600-1000 upregulated genes). Our data (Fig. R1) shows that OAS1 and Mx1 are induced to similar levels by type I and III IFNs, and to lower levels by type II IFN. However, type II IFN induced the highest levels of CXCL10. This is in line with the well-known notion that different types of IFNs induce distinct but overlapping transcriptional programs (e.g. Schneider et al, Annual review of immunology; Schoggins et al, Current opinion on Virology, 2011; Liu et al, PNAS, 2012; Pervolaraki et al, PLOS Pathogens, 2018; Schoggins et al, Annual Review of Virology, 2019). This agrees with our previous data showing that type II and type III IFNs restrict SARS-CoV-2 in a synergistic manner (Hayn et al, Cell Reports, 2021). However, rather than abundance, but more so the individual impact of an ISG on SARS-CoV-2 is important (compare Nchioua et al, mBio, 2020). Thus, analysis of differential ISG induction by IFNs is not suitable to explain observed differences of IFNs on SARS-CoV-2 replication or state how effective different IFNs are in activating antiviral programs.

Fig. R1: Transcriptional induction of the ISGs OAS1, Mx1 and CXCL10 in Calu-3 cells 18 h post treatment with indicated IFNs (500 IU/mL, IFNλ1: 100 ng/mL) as assessed by qRT-PCR. N=3±SEM.

March 20, 2023

RE: Life Science Alliance Manuscript #LSA-2022-01745-TRR

Dr. Konstantin M.J. Sparrer
University Hospital Ulm
Meyerhofstr.1
Ulm, BW 89081
Germany

Dear Dr. Sparrer,

Thank you for submitting your Research Article entitled "Reduced replication but increased interferon resistance of SARS-CoV-2 Omicron BA.1". It is a pleasure to let you know that your manuscript is now accepted for publication in Life Science Alliance. Congratulations on this interesting work.

DISTRIBUTION OF MATERIALS:

Again, congratulations on a very nice paper. I hope you found the review process to be constructive and are pleased with how the manuscript was handled editorially. We look forward to future exciting submissions from your lab.

Sincerely,
